# Production, Processing, and Protection of Microalgal n-3 PUFA-Rich Oil

**DOI:** 10.3390/foods11091215

**Published:** 2022-04-22

**Authors:** Xiang Ren, Yanjun Liu, Chao Fan, Hao Hong, Wenzhong Wu, Wei Zhang, Yanwen Wang

**Affiliations:** 1INNOBIO Corporation Limited, No. 49, DDA, Dalian 116600, China; liuyj@innobio.cn (Y.L.); fanc@innobio.cn (C.F.); hongh@innobio.cn (H.H.); wu@innobio.cn (W.W.); 2DeOxiTech Consulting, 30 Cloverfield Court, Dartmouth, NS B2W 0B3, Canada; wzhang14@gmail.com; 3Aquatic and Crop Resource Development Research Centre, National Research Council of Canada, 550 University Avenue, Charlottetown, PE C1A 4P3, Canada

**Keywords:** microalgae, n-3 PUFA, EPA and DHA, biosynthesis pathways, heterotrophic cultivation, micronutrients, lipid purification, microencapsulation

## Abstract

Microalgae have been increasingly considered as a sustainable “biofactory” with huge potentials to fill up the current and future shortages of food and nutrition. They have become an economically and technologically viable solution to produce a great diversity of high-value bioactive compounds, including n-3 polyunsaturated fatty acids (PUFA). The n-3 PUFA, especially eicosapentaenoic acid (EPA) and docosahexaenoic acid (DHA), possess an array of biological activities and positively affect a number of diseases, including cardiovascular and neurodegenerative disorders. As such, the global market of n-3 PUFA has been increasing at a fast pace in the past two decades. Nowadays, the supply of n-3 PUFA is facing serious challenges as a result of global warming and maximal/over marine fisheries catches. Although increasing rapidly in recent years, aquaculture as an alternative source of n-3 PUFA appears insufficient to meet the fast increase in consumption and market demand. Therefore, the cultivation of microalgae stands out as a potential solution to meet the shortages of the n-3 PUFA market and provides unique fatty acids for the special groups of the population. This review focuses on the biosynthesis pathways and recombinant engineering approaches that can be used to enhance the production of n-3 PUFA, the impact of environmental conditions in heterotrophic cultivation on n-3 PUFA production, and the technologies that have been applied in the food industry to extract and purify oil in microalgae and protect n-3 PUFA from oxidation.

## 1. Introduction

Microalgae are a group of multilineage and highly diverse microorganisms, ranging from prokaryotic cyanobacteria to eukaryotic single-celled organisms, which grow in freshwater and saltwater [1,2]. Additionally, these microorganisms can carry out photosynthesis, while some species are heterotrophic and can grow in an organic matrix in a contained fermentation process. Microalgal diversity across the planet is conservative and estimated to have over hundreds of thousands of species, but only tens of thousands have been classified and about a hundred species have been fully studied [3,4,5]. Although limited, available information shows that many high-value biomolecules are produced by microalgae, including proteins, n-3 polyunsaturated fatty acids (PUFA), polysaccharides, vitamins, pigments, and antioxidants. Among them, n-3 PUFA have been of great interest because of their various nutritional and physiological values and health benefits, such as antioxidant, antihypertensive, anti-inflammatory, immune regulation, antiviral, liver protection, neuroprotection, cardiovascular prevention, cholesterol reduction and anticancer [6]. The development of large-scale technologies for microalgae cultivation dates back to the 1960s [7]. Nowadays, microalgae have already become one of the main biological systems of n-3 PUFA production and the main source of n-3 PUFA for vegetarians. Compared with terrestrial plants, microalgae have several superior characteristics, such as high yield, short production cycle, free irrigation land, flexible metabolism, environmental protection, etc. and thus have been recognized as “strategic crops” [8]. Some microalgal species have been reported to produce lipids in quantities as high as 70% of their dry matter [9]. The advantages of microalgae become more relevant today than ever before as the supply of n-3 PUFA has encountered serious challenges as a result of an increase in the global population, climate change, overfishing of marine species, and insufficient production of aquaculture [10]. The current review focuses on (1) various biosynthesis pathways and recombinant engineering approaches that can be used to manipulate microalgal growth and produce specific n-3 PUFA, (2) growth conditions in heterotrophic cultivation and their influences on n-3 PUFA production, (3) extraction and purification technologies, and (4) technologies that have been applied at industrial scales to produce, process, and protect n-3 PUFA, particularly EPA and DHA in food and dietary supplement industries.

## 2. Literature Search and Analysis

A comprehensive literature search was conducted by accessing several databases, such as PubMed, ISI-Medline, and Google Scholar. The keywords were chosen based on the key objectives of this review paper that are reflected in the title, subtitles and contexts. The published papers including patents in the past 20 years have been mainly selected, mirroring the time window for the knowledge creation, technology development, industrial production and commercialization of microalgal n-3 PUFA products. A total of 352 references have been included in this review. In each section, the selected references were summarized and analyzed to support our statements or help to explain the differences or discrepancies between different studies. We also attempted to draw our conclusions and point out the strengths and weaknesses, and accordingly, the areas or directions that need to be addressed in future research and technology/product developments.

## 3. Characteristics of Microalgal Polyunsaturated Fatty Acids

A number of microalgal species are typically rich in fatty acids, waxes, sterols, short- and long-chain hydrocarbons, pigments and many other biomolecules that are valuable for human nutrition and health. One group of the most prominent biomolecules are the PUFA-rich lipids [11,12]. There are two classes of PUFA, n-3 or n-6, depending on the position of the first double bond from the methyl end [13,14]. n-3 PUFA are characterized by the presence of the first double bond located at the third carbon from the methyl group, including hexadecatrienoic acid (HTA), eicosatetraenoic acid (ETA), α-linolenic acid (ALA), eicosapentaenoic acid (EPA), docosapentaenoic acid (DPA), and docosahexaenoic acid (DHA) [15]. ALA is the precursor of DHA and EPA, which are very important to human physiological functions and health [14,16,17]. Different from n-3 PUFA, n-6 PUFA have the first double bond positioned at the sixth carbon from the methyl group, including linoleic acid (LA), γ-linolenic acid (GLA), and arachidonic acid (ARA) [18]. LA is the essential n-6 fatty acid and the precursor of ARA and other longer chain n-6 PUFA. Although ALA can be converted into longer chain n-3 PUFA in the human body, the efficiency is low and thus the direct intake and/or supplementation of EPA and DHA have been well recognized as an important approach to meet the nutritional requirements and maintain health.

In addition to marine sources such as fish, microalgae have been emerging as cost-effective sources of n-3 PUFA, particularly EPA and DHA [19]. Some microalgae accumulate high amounts of lipids, ranging from 20% to 70% of the total dry biomass [9,20,21]. *Chlorella*, *Spirulina*, *Porphyridium cruentum*, *Phaeodactylum tricornutum*, *Pavlova lutheri*, and *Arthrospira platensis* have been used to produce novel dietary and pharmaceutical lipid supplements due to their high contents of n-3 PUFA [22,23]. In addition, microalgae represent the primary sources of essential fatty acids in food webs for zooplankton, fish, and other multicellular organisms [24,25]. Fish are rich in both EPA and DHA as a result of food web magnification of DHA from flagellates and EPA from diatoms [22,24]. In fact, microalgal lipids have become an effective substitute of fish oil in foods and dietary supplements, providing the essential fatty acids (especially n-3 EPA and DHA) in humans without the drawbacks associated with fish oils, such as a fishy smell, unsustainability, and non-vegetarian nature [26].

Due to their advantages in production, environmental footprint and downstream utilization, microalgae have been regarded as promising sources of n-3 PUFA. These microorganisms are increasingly cultivated at industrial scales for the production of n-3 PUFA, which is used alone or to enrich fish oil with DHA and/or EPA in a number of food and dietary applications [1,3,11]. n-3 EPA and DHA are currently used mainly in food and dietary supplement products, especially infant formula due to limited supply [27]. However, with the ageing of the global population and increasing awareness of the health benefits of n-3 PUFA, the consumption of n-3 PUFA has been steadily increasing in the past few decades and will continuously rise [28,29,30]. Accordingly, the increase of n-3 PUFA production using microalgae by discovering new species, developing and applying new or improved production and processing technologies has been emerging as a strategic approach [1,31,32,33]. 

### 3.1. Microalgal Production of n-3 PUFA

#### 3.1.1. Production of n-3 PUFA in Wild-Type Microalgae

Microalgae possess the potential to synthesize and accumulate a high amount of PUFA in comparison to other biology systems that produce edible oils [12]. Generally, the carbon chain length of fatty acids in natural microalgae is between C16 and C22; however, some species can synthesize very-long-chain fatty acids, such as those with 24 carbons [34]. There is a large range in the content of PUFA in microalgae. It is reported that *C. cohnii* accumulated 25–60% DHA [35], and EPA in *Nannochloropsis oculata* accounted for 49% of the total fatty acids [36,37]. The fatty acid composition of cyanobacteria is generally simpler and predominated by those with 16 and 18 carbons, of which the PUFA mainly include LA (18:2Δ9,12), ALA (18:3Δ9,12,15), and GLA (18:3Δ6,9,12) [38]. Chlorophytes (Plantae) contain a high amount of PUFA with 18 carbons, while rhodophytes and glaucophytes are rich in PUFA with 20 carbons, with ARA and EPA being exemplified in a study on *Porphyridium cruentum* (*Porphyridiophyceae*) [39]. Other studies have shown that *Xanthophyceae* and *Eustigmatophyceae* (*Chromista—Ochrophyta*) are rich sources of ARA and EPA [40,41]. A small amount (less than 1%) of very-long-chain PUFA C28:7(n-6) and C28:8(n-3) were found in the toxic dinoflagellate *Karenia brevis* (*Dinophyceae*) [41,42,43]. Diatoms contain high contents of long-chain unsaturated fatty acids and have attracted the greatest attention in fatty acid research and product development [44,45,46]. The fatty acid composition is principally associated with the distribution of lipid classes. In general, fatty acids C14:0, C16:0, and C16:1 are associated with triacylglycerols, while EPA and DHA are associated with polar lipids [47,48,49,50]. The fatty acid composition in some microalgal species is summarized in Table 1 [12,51].

#### 3.1.2. Production of PUFA in Recombinant Microalgae

The development of transcriptomic and proteomic platforms coupled with advances in novel genetic techniques, such as CRISPR/Cas9, have nowadays provided important tools in understanding and selecting microalgae species/strains for the production of target compounds such as n-3 PUFA, especially EPA and DHA [61]. The production of target fatty acids in microalgae can be enhanced using genetic manipulations that alter the metabolic pathways of lipid anabolism and catabolism [62]. Whole genomes are available in over 20 microalgal species, and some molecular tools can be used to improve lipid production [63]. A few studies led to the selection of species *Phaeodactylum tricornutum* and *Schizochytrium* sp. for improved triacylglycerol yield [64]. One genetic approach is the target knockdown of a key gene involved in triacylglycerol catabolism [65] or by disabling the competitive carbon metabolic pathways with starchless mutants [66]. Another approach is the transfer of novel fatty acid biosynthesis genes from other microalgae or microorganisms, with the directed gene up-regulation strategy being employed more frequently. By expressing Δ5-elongase from *Ostreococcus tauri* and a glucose transporter from the moss *Physcomitrella patens*, *Phaeodactylum tricornutum* produces 36.5% DHA and 23.6% EPA of the total fatty acids, making the technology highly attractive in the commercial production of n-3 EPA and DHA [67]. Exchanging the acyltransferase gene via deletion and replacement of the native acyltransferase with its homologue, *Shewanella* sp. produces DHA and EPA at levels up to 28.8 and 2.3 g/L [68]. Disrupting the expression of the fatty acid synthase gene combined with the overexpression of acetyl-CoA carboxylase and diacylglycerol acyltransferase in Thraustochytrid *Aurantiochytrium* resulted in a high yield of DHA, 61% of the total fatty acids [69].

Many studies have been conducted to improve the productivity of specific biomolecules in microalgae through recombinant DNA and various genetic techniques. To date, a few recombinant products have been successfully produced in microalgae, with the majority being achieved in *Chlamydomonas reinhardtii* [11,70,71,72,73]. Moreover, these techniques are predominantly at the laboratory stage or scale; few recombinant microalga products are produced and marketed commercially. It is apparent that research and product development in microalgae using recombinant techniques are still at early stages and more work is warranted.

### 3.2. Synthetic Pathways of PUFA in Microalgae

Microalgae are highly diversified biological systems, accounting for 40% of global photosynthesis. Because of their high contents of PUFA and wide adaptations to environmental factors, they are considered to have great commercial potential [74]. Subsequently, understanding the utilization and transformation of solar energy by microalgae becomes a key element in studying microalgal lipid metabolism, which is of great benefit to the industrial production of high value n-3 EPA and DHA [20].

In microalgae, sufficient acetyl-CoA should be generated or provided to maximize PUFA synthesis. Glucose or carbon dioxide (CO_2_) can be used as the carbon source, which distinguishes autotrophic, mixotrophic, and heterotrophic microalgae [75]. Glucose can be transformed into acetyl-CoA via glycolysis, while CO_2_ needs to be fixed through the Calvin cycle before glycolysis. Acetyl-CoA in microalgae has two possible fates: participating in tricarboxylic acid (TCA) cycle or transforming to malonyl-CoA for fatty acid synthesis. When grown in nitrogen-rich media, microalgal cells allocate more acetyl-CoA to the TCA cycle where an intermediate α-ketoglutarate is an essential substrate for nitrogen assimilation [76]. As such, the nitrogen-deficient medium is more favorable to fatty acid synthesis, where acetyl CoA carboxylase catalyzes the formation of malonyl-CoA from acetyl-CoA [17,61]. 

There are two fundamentally different pathways, aerobic and anaerobic, to synthesize PUFA in microalgae (Figure 1). The aerobic pathway is catalyzed by alternating desaturases and elongases, with multiple desaturations and extensions. Acyl carrier protein (ACP) transacylase transfers the malonyl group from malonyl-CoA to malonyl-ACP. Acyl-ACP is the carbon source for chain elongation. This reaction is catalyzed by ketoacyl-ACP synthases (KASIII, KASI, and KASII). After each condensation, a reduction, dehydration, and second reduction occur [77]. The synthesized fatty acids undergo chain elongation and unsaturation in the presence of elongases and desaturases, respectively, and are then transported into the cell cytoplasm where triacylglycerols are assembled. Regarding the synthesis of PUFA for example, EPA a palmitic acid is synthesized and elongated to a stearic acid by adding two carbon atoms to the main chain through a reaction catalyzed by an elongase [1]. After the introduction of two double bonds with the first double bond at the third carbon from the methyl group, the stearic acid becomes ALA, and this process involves a reaction catalyzed by stearoyl ACP desaturase; ALA is further transformed into EPA through processes catalyzed by elongase and desaturase [78]. The second mechanism of fatty acid synthesis in microalgae is the anaerobic pathway involving polyketide synthase (PKS pathway), which requires fewer reducing equivalents and produces specific fatty acids, such as DPA and DHA. The PKS pathway involves seven proteins, which are 3-ketoacyl synthase, 3-ketoacyl-ACP-reductase, dehydrase, enoyl reductase, dehydratase/2-*trans* 3-*cis* isomerase, dehydratase/2-*trans*, and 2-*cis* isomerase, with the addition of two carbons and/or a double bond. Acetyl-CoA and malonyl-CoA as the primary building blocks do not require in situ reduction of the intermediate because oxygen is not involved in double-bond generation. This pathway occurs in *Schizochytrium* sp. and other *Thraustochytrid* organisms [75,76,79].

## 4. Commercial Production of PUFA Using Microalgae

### 4.1. Environmental Factors Influencing Microalgal PUFA Production

Microalgae interact with environmental factors to regulate the synthesis and accumulation of various biomolecules, including fatty acids. These factors mainly include light, temperature, salinity, and nutrients. In terms of nutrients, carbon, nitrogen, and phosphorus are the most important ones, while other minerals also need to be optimal in order to obtain a high yield and a favorable fatty acid profile of lipids. 

#### 4.1.1. Light

It is well understood that light is essential to photosynthetic organisms [80,81] and plays a key role in microalgae growth [82]. The wavelength, intensity and duration all influence the synthesis and accumulation of PUFA [83,84]. Generally, a high light intensity results in a low content of PUFA in microalgae biomass. In *Nannochloropsis* sp., the degree of unsaturation of fatty acids decreases with the increase in irradiance, with a threefold decrease in the percentage of total n-3 fatty acids (from 29% down to 8% of the total fatty acids), caused mainly by a decrease in EPA content [85]. A study demonstrated that in *Chlorella vulgaris*, the increase in light intensity from 37.7 to 100.0 μmol m^−2^ s^−1^ resulted in a decrease in PUFA from 27.4% to 21.7% of the total fatty acids, especially EPA and DHA, which were decreased by 70% and 50%, respectively [86]. In *S. piluliferum*, a high light intensity resulted in a decrease in almost all fatty acids [87]. The observed effects of light on PUFA synthesis and accumulation in microalgae can be partly attributed to the generation of intracellular reactive oxygen species (ROS), which increase PUFA oxidation [1]. As the other extremity of light intensity, the effect of darkness on the lipid content and fatty acid composition has also been studied [80,88]. For example, in green algae *S. capricornutum*, dark treatment decreased the oleate content but increased the linoleate content [89].

In addition to light intensity, light wavelength also affects the synthesis and accumulation of PUFA in algal cells [90]. In a study, *Chlorella vulgaris*, *Chlorella pyrenoidosa*, *Scenedesmus quadricauda* and *Scenedesmus obliquus* were cultivated under light with different wavelengths, and it was discovered that blue light was much more favorable to the accumulation of LA in all four algal strains than red light [91]. It was also observed that the ratios of n-6/n-3 PUFA in *Chlorella* and *Scenedesmus* cultured under blue light were much lower than those grown under red light [92,93]. It was reported that ultraviolet light induced the biosynthesis of PUFA in the acidophilic microalga *Coccomyxa onubensis*, and the efficacy was dependent on the sensitivity or tolerance of microalgae [94]. The reductions of EPA and DHA by ultraviolet were more sensitive in nutrient-deprived cells [95]. It is apparent that the light wavelength, intensity, and duration substantially affect the synthesis and accumulation of PUFA in microalgal cells, and optimization is critical to the maximal production of n-3 PUFA, particularly EPA and DHA. 

#### 4.1.2. Temperature

Temperature is another important environmental factor that influences microalgae growth, lipid content and fatty acid composition [80,96]. Temperature can be used as a stressor to encourage the production of valuable metabolites and improve the content and profile of PUFA in microalgae lipids [92,93,97,98]. It was found that *Leptocylindrus danicus* grown at 14 °C yielded higher levels of PUFA than those grown at 26 °C [93]. The positive effect of low temperatures on PUFA synthesis was also observed in other species, such as *Nannochloropsis salina*, *Isochrysis galbana*, *Rhodomonas salina*, and *Dixioniella grisea* [93,97,99].

In general, higher temperatures favor cell growth while lower temperatures favor fatty acid synthesis in microalgae [100,101]. PUFA are functional elements of algal membrane lipids in the forms of phospholipids and glycolipids and contribute to cell-signaling and physiological functions. Cultivation under low temperature conditions promotes the accretion of PUFA in the cell membrane, which increases membrane fluidity [102], while cultivation at high temperatures leads to the opposite effects [103]. The analysis of lipids and fatty acids of microalgae cultivated under different temperatures revealed that total lipids accumulated at a higher rate at 30 °C with a slight decrease in the proportion of non-polar lipids, while algae grown at 15 °C had higher contents of ALA and DHA but lower amounts of monounsaturated and saturated fatty acids [104]. The impact of temperature on lipid accumulation and fatty acid profiles is consistent with observations in marine species including fish. This knowledge is highly valuable to algal cultivation targeting the production of n-3 PUFA, EPA and DHA.

#### 4.1.3. Salinity

Salinity in a cultivation system also influences the fatty acid profile of algal lipids [100]. For the cultivation of marine microalgae, it is necessary to simulate the salinity of the ocean, but it is also easy to corrode the fermentation tank [1]. In freshwater microalgae cultivation, a high salinity level often has a negative effect on the accumulation of PUFA [105]. A decrease in PUFA content with increasing salinity levels was observed in *Chlamydomonas reinhardtii* grown in media supplemented with 0.1, 0.2, and 0.3 M NaCl, and higher salinity (0.2 and 0.3 M) was even fatal to microalgae [106]. It was reported that the PUFA content in *Desmodesmus abundans* grown in media supplemented with 20 g/L NaCl was much lower than the control without salt supplementation [107]. It was observed that NaCl induced an increase in ROS in *Chlorococcum humicola* and *Chlorella vulgaris*, resulting in the increased oxidation of algal PUFA and thus lower PUFA contents [108,109]. Similar results were observed in *Microcystis aeruginosa* [110]. Contrarily, a few studies showed that salt stress may promote lipid accumulation in other microalgae [111]. For example, in *Dunaliella salina*, cultivation at low-to-high salt concentrations (0.5–3.5 M NaCl) resulted in an increased expression of β-ketoacyl-coenzyme A (CoA) synthase (Kcs), which catalyzes the first and the rate-limiting step of fatty acid chain extension [112]. The salt-induced Kcs, jointly with fatty acid desaturases, was thought to change fatty acid synthesis and play an important role in the adaptation of the intracellular membrane compartments, resulting in high internal glycerol concentrations to balance the external osmotic pressure created by high salinity levels [112]. Preventing over-oxidation of lipids in algal cells requires a strict control of salinity levels in culture media. This can be achieved by determining the correlation between the salinity levels and total lipid and PUFA contents in a given microalgal species or strain.

#### 4.1.4. Carbon

Carbon, as a macronutrient [113], is critical to the biosynthesis of PUFA and lipids in microalgae cells, and the sources include inorganic and organic carbons [114,115,116]. The inorganic carbon required for photosynthesis can be normally obtained from the atmospheric CO_2_ or from dissolved bicarbonate ions in the media [113,117]. The concentration of inorganic carbon has a significant effect on the composition and content of fatty acids [60,118]. It was observed in *Pavlova lutheri* that when bicarbonate concentration in the medium increased from 2 to 8 mM, the percentage of PUFA in total fatty acids increased by 5.6%, while the total fatty acids increased by 0.8 pg/cell [119]. In *Chlorella kessleri*, more ALA was produced when they were cultivated in media containing lower levels of CO_2_ [120]. It was observed that unsaturation levels were higher in microalgae cultured at lower CO_2_ concentrations, which was attributed at least in part to the repressed fatty acid synthesis, allowing desaturation of the pre-existing fatty acids [120]. The effect of CO_2_ concentration on the content and composition in chloroplast lipids and whole cell lipids has also been investigated in a unicellular halophilic green alga *Dunaliella salina*, which is known to be susceptible to CO_2_ stress [121]. It was found in this study that even a one-day-long increase in medium CO_2_ concentration from 2% to 10% provoked an increase in total fatty acids by 30%. The results of this study indicate that high CO_2_ concentrations increase fatty acid synthesis de novo but inhibit fatty acid elongation and desaturation. These changes might represent an adaptation process and mechanism to ensure effective photosynthesis of microalgae in environments with different CO_2_ levels.

Organic carbon sources are relatively expensive if used for microalgae fermentation [100]. Glucose is commonly used in commercial production and usually comes from glucose syrup produced by amylase conversion of cereal or potato starches. The concentration of glucose in the fermentation media is in a range of 5–40 g/L [122]. Although glycerol is a potential substitute of glucose, it is often associated with chemical impurities [123]. In microalgae, both glucose and glycerol are first converted to pyruvic acid then to acetyl CoA via the citric acid cycle, serving as the basic carbon source for lipid production [100]. Since the metabolic pathway for the conversion of glucose to acetyl-CoA in algal cell is much shorter than that of CO_2_ to acetyl-CoA, glucose might be more favorable to the synthesis of lipids and PUFA than CO_2_ in microalgae [1]. This explains why the heterotrophic mode of organic carbon source is mainly used commercially to achieve high yields of lipids and PUFA in microalgae [124]. Different carbon sources can be selected for the production of different compounds using microalgae cultivation and fermentation. From the perspective of environmental protection and cost savings, inorganic carbon sources should be considered for biofuel production, while organic carbon sources may be used for the production of higher value products, such as PUFA. It is evident that the selection of carbon source in microalgae cultivation and fermentation depend on the final products of interest and production systems employed [125,126].

#### 4.1.5. Nitrogen

Similar to carbon, nitrogen for microalgae is also divided into inorganic (nitrate, nitrite, ammonia, and molecular nitrogen) and organic (urea, degraded proteins such as yeast extracts and soya peptones) sources [127,128,129]. Although inorganic nitrogen sources are rich and economical, they lack trace minerals and other nutrients, such as vitamins. Organic nitrogen sources are more nutritious but more expensive [100]. A nitrogen source is important during the first phase of algal fermentation or biomass development, where it is used for amino acid biosynthesis. Sufficient nitrogen supply makes microalgae cells grow and accumulate biomass rapidly. When the nitrogen supply is exhausted in the fermentation media, microalgae shift their metabolisms to fatty acid synthesis from available carbon sources [3]. Photosynthesis decreased substantially under the condition of nitrogen limitation because a smaller proportion of nitrogen-rich components were synthesized with respect to energy storage compounds [90]. Nevertheless, studies on different microalgal species showed that most microalgae increased lipids and PUFA productions when grown in nitrogen-deficient media [130,131]. *P. tricornutum*, *Nannochloropsis* sp. and *Chlorella pyrenoidosa* have were examined in multi-laboratories under nitrogen starvation and increases in lipid content were observed [44,79,132,133,134]. Similar phenomena were seen by others in *Phaeodactylum tricornutum*, *Pavlova viridis*, and *Tetraselmis subcordiformis* [135,136]. During nitrogen starvation, the accumulation of triacylglycerols increased in *Chlorella pyrenoidosa*, paralleled by the increase in the gene expression of acetyl-CoA carboxylase and diacylglycerol-*O*-acyltransferase [124,134,137]. A study examined the effect of nitrogen starvation on the lipid production and fatty acid profile in three microalgal species cultured at nitrogen concentrations between 0 and 1.76 mmol/L, and found that all three microalgae had the highest lipid accumulation when cultured at a nitrogen concentration of 0.22 mmol/L [136]. As such, nitrogen starvation is commonly used to induce PUFA accumulation in microalgae cultivation. However, nitrogen starvation reduces the photosynthetic activity and growth rate [64]. Therefore, it is crucial to adjust the cultivation strategy and make a trade-off between PUFA content and biomass production rate in order to achieve a maximal yield of target fatty acids such as n-3 EPA and DHA. 

#### 4.1.6. Phosphorus

In microalgae, phosphorus is assimilated mainly as phosphate ions, with the ability of uptaking organic phosphorus as well [138]. Riekhof et al. showed that *C. reinhardtii* reduced the concentration of phosphoglycerides when cultured under a condition of phosphorusstarvation [139]. In the freshwater eustigmatophyte for example, *Monodus subterraneus*, changes in phosphate concentration caused significant changes in the fatty acid and lipid compositions. As the phosphate concentration decreased from 175 to 52.5, 17.5 and 0 μM (K_2_HPO_4_), the percentage of PUFA and EPA in the total fatty acids decreased [140]. Phospholipid synthesis is particularly affected in the phosphorus-limited culture, thus leading to the increased triacylglycerol accumulation [141,142,143]. Phosphorus-limitation in the media led to increased lipid contents in *P. tricornutum*, *Chaetoceros* sp. and *P. lutheri* but decreased lipid contents in *Nannochloris atomus* and *Tetraselmis* sp., and furthermore, severe phosphorusstarvation resulted in higher contents of palmitate and oleate but lower contents of stearidonic acid, EPA and DHA [144]. An increased level of PUFA in all the individual lipids analyzed(phosphatidylcholine, phosphatidylglycerol, digalactosyldiacylglycerol, monogalactosyldiacylglycerol, sulphoquinovosyldiacylglycerol) has been reported in phosphorus-starved cells of the green alga *C. kessleri* [145,146]. It has been recognized that phosphatelimitation causes the replacement of membrane phospholipids with non-phosphorus glycolipids and betaine lipids, an effective phosphate-conserving mechanism in microalgae [147]. This natural adaptation to poor phosphorus environments is termed luxury phosphorus uptake [148]. A decrease in phosphorus concentration changes the way of lipid synthesis, metabolism, and accumulation, similar to the effect of nitrogen but to different extents [113].

#### 4.1.7. Other Minerals

Micronutrients, needed in trace amounts (such as Co, Cu, Fe, Mg, Mn, Mo, and Zn), also influence algal growth as well as lipid synthesis and metabolism since they can alter enzyme activity [19]. However, a high concentration of trace elements is toxic [149,150]. Iron and zinc have been shown to influence triacylglycerol accumulation in *Chlamydomonas* [151,152]. In iron-starved *C. reinhardtii*, lipid droplets and triacylglycerols were accumulated and an increased saturation index was noted, suggesting that the desaturase activity was compromised [151,153]. Like iron, copper is needed for certain enzyme activities [80,154]. Magnesium, an important component of the photosynthetic apparatus, is also critical for microalgae growth and biomass production [155,156]. Exposure to heavy metals (Cu^2+^, Zn^2+^ and Cd^2+^) led to an increase in oleate and altered the percentage of linoleic acid and stearidonic acid in *S. capricornutum*, and treatment with these minerals also significantly increased fatty acid desaturation [120]. Research on the effect of microelements on the growth and fatty acid profile has fallen behind, and more work is required to understand the roles of these nutrients in algal growth and biosynthesis of fatty acids in microalgae. 

### 4.2. Commercial Cultivation Systems for Microalgal PUFA Production

The growing interest in microalgae has led to a diversification of cultivation systems tailored to different microalgal species and biomolecules [12]. Accordingly, various methods and technologies have been created for cultivating and processing microalgae for food, feed, biofuels, and other high-value biomolecules [157,158,159,160]. Open ponds and closed photobioreactors for autotrophic cultivation along with industrial fermenters for heterotrophic production play important roles in lipid and PUFA production using microalgae [12]. To date, the production of high-value PUFA using microalgae is becoming available commercially, whereas developing more cost-effective technologies in both cultivation and downstream processing remain as continued challenges.

#### 4.2.1. Photoautotrophic Cultivation

Presently, photoautotrophic cultivation is considered as the most frequently used tactic for growing microalgae [161], which mainly includes photobioreactors and open ponds [162]. Among them, open-pond cultivation is the oldest, most widely used, and least complicated system. The development of this technology dates back to the 1950s [163]. A large open system mainly consists of a raceway pond with paddle wheels (a shallow basin lower than 0.3 m) [164,165]. This traditional method has some drawbacks, such as low productivity, difficulty controlling the growth conditions due to seasonal changes, easy pollution by microorganisms, and insufficient light caused by mechanical stirring [12,113,166]. However, because of its low energy consumption, simple process, and easy maintenance, open-pond cultivation is considered to be an economical system [167,168,169].

From the 1980s to 1990s, a closed autotrophic photobioreactor system remained as the research hotspot of microalgae biotechnology and for the industrial production of microalgae biomass and biomolecules of interest [170,171,172]. Compared with an open-pond system, a closed photobioreactor has several advantages, such as smaller land requirement, control of operational/growth conditions, larger surface area, high nutrient utilization rate, and a closed environment, allowing aseptic cultures [173,174,175]. The close photo bioreactoris also adaptable to different growing conditions required for different microalgae species, even delicate species, and produce a higher purity of target products [176,177]. However, culturing microalgae in a photobioreactor system also has some disadvantages, such as high construction and operation costs and oxygen accumulation resulting from photosynthesis. In addition, light limitation is the main factor affecting autotrophic productivity, and the illumination in photobioreactor is not uniform. Light penetration or intensity decreases rapidly as a result of geometric structure, system hydrodynamic properties, cell auto-shading and the formation of biofilms [177]. In order to reduce the impact of the obstacles attached to the traditional photoautotrophic technology, researchers have been actively exploring new methods, such as the use of a scalable membrane photobioreactor (SM-PBR), which was proposed to realize high-efficiency nutrient recovery for microalgal lipid production [178]. A study showed that microalgae in the traditional photobioreactor (T-PBR) died in two days, while microalgae in the SM-PBR grew well, with the biomass concentration increasing from 0.10 g/L to the maximum of 2.13 g/L in ten days [179]. In addition, an outdoor photobioreactor using fed-batch cultivation technology has been reported. In this study, the microalgae *Scenedesmus abundans* were cultivated in five identical airlift photobioreactors in batch and fed-batch modes. It was found that the fed-batch mode produced higher amounts of biomass and lipids (ALA up to 14% (*w*/*w*) of the total fatty acids) in harsh outdoor conditions [180]. A novel pyramid photo-bioreactor (PBR) was created, which is a modified version of flat-plate PBR and consisted of four completely separated equal-volume chambers. This system uses both external and internal light sources to improve the control of light intensity and light homogeneity, and can thus be used to better manage the production rate of target biomolecules or biomass of a given microalgae species [181].

#### 4.2.2. Heterotrophic Fermentation

Heterotrophic fermentation is another microalgae culture technology commonly used in the industrial production in which the energy is provided by organic matter [182,183]. Many microalgal species can grow on organic carbon sources such as glucose, which can be easily absorbed and converted to acetyl-CoA and greatly improve the yield of lipids and PUFA. At present, most vegetarians in the world obtain n-3 PUFA from microalgae, and the production is carried out in heterotrophic fermentation tanks. *Dinophyc**e**ae*, *Schizochytrium* sp. and *Thrustochytriaceae* are the most commonly cultured and have a lipid content of 40–60%. The lipid content of *Aurantiochytrium limacinum SR2**1* can reach up to 84% when cultured with glucose or glycerol as a carbon source [184]. After the optimization of growth conditions, a high content of PUFA can be produced in microalgae [185], which can be used as an eccentric source of DHA and EPA in foods and dietary supplements.

Heterotrophic culture of microalgae can significantly increase cell mass and total lipids [161,186]. Compared to photo autotrophy, biomass concentration can be increased to 18 g L^−1^ [187] and 24 g L^−1^ [188], which is 30–50% of the total lipids (50 or 60 g L^−1^) produced by the traditional industrial yeast fermentation [189,190]. Compared with the autotrophic fermentation, heterotrophic fermentation accumulates 3–4 times more lipids. Similarly, the lipid production rate of *Scenedesmus obliquus* in mixotrophic cultivation (11.6–58.6%) was higher than that in the photoautotrophic cultivation (7.14%) [191]. The mixotrophic cultivation of *Chlorella vulgaris* produced biomass that was 4.43 times higher than the photoautotrophic cultivation [192]. However, the production cost of this culture system is very high due to the cost for maintaining sterile conditions, in addition to the costs of organic carbon source and oxygen supply. For example, the production of 1 ton of microalgae in a heterotrophic cultivation system requires up to 5 tons of sugar [3].

Microalgae biofilm combines the advantages of heterotrophic fermentation (high biomass yield) and autotrophic photobioreactor (sunlight as energy source and CO_2_ as carbon source) [193,194]. Biofilms can be formed with different microorganisms (bacteria, yeasts, microalgae), which are usually known for their negative effects of biological pollution and drug resistance in many applications [194]. In recent years, researchers have found that many biofilm cultivation systems show great potential, such as constantly or intermittently submerged and perfused systems [195]. Biofilm cultivation not only has a higher productivity but can also be harvested by simple scraping, which can greatly reduce energy consumption [196,197].

### 4.3. Harvesting and Drying of Microalgae

Harvesting is the solid–liquid separation of microalgae cells from the growth media. Because the concentration of microalgae biomass in the culture medium is very low, it is impossible to perform further downstream treatments, such as cell fragmentation and oil extraction and fractionation [113]. The typically low biomass yield in microalgae production systems imposes severe economic and energetic restrictions to harvesting and subsequent biorefinery [12]. Therefore, the harvest of microalgae cells is the key prerequisite operational step for the production of microalgae oil. Preconcentration and dehydration are the main collection methods of microalgae cells. Preconcentration of microalgal cells can be accomplished by several methods such as flocculation, centrifugation, sedimentation and filtration, aiming to increase the initial biomass content from 0.5–1.0% to around 3% [198]. Microalgae generally carry negative charges and require positive-ion-containing flocculants such as iron chloride and aluminum sulphate for coagulating the biomass [199]. For larger sized filamentous microalgae such as *A. platensis*, the biomass can be efficiently harvested using vibrating sieves, a relatively simple technology [200]. The representative harvesting methods of microalgae are summarized in Table 2.

Oil is often extracted from the dry matter of microalgae [212]. To obtain dry matter, the harvested microalgae is first concentrated by a dewatering process to contain around 25% biomass [198]. Then, different drying technologies have been used to dry the concentrated microalgal pastes, including spray drying, drum drying, freeze drying, sun drying, and oven drying. Nevertheless, each method possesses its own pros and cons [213]. Freeze drying, also known as lyophilization or cryodesiccation, has been found to be an efficient but costly method, and oven drying consumes even more energy [12]. The choice of harvesting and drying methods is dependent on multiple factors, for example, the species of microalgae, cultivation time, maintenance and suitability of cultures for commercialization [214]. To overcome the negative repercussions related to the conventional methods, there is a large demand to develop new methods and technologies for cell harvesting and drying [215,216,217].

### 4.4. Pretreatment of Microalgae by Cell Wall Disruption

The application of cell wall disruption in the extraction of microalgal oil depends on the cell wall structure, which varies greatly among species [215,218,219,220]. Cell disruption can be achieved by physical (ultrasonication, high-pressure homogenization, bead milling, cryogenic grinding, and pulsed electric field), chemical (acid, alkaline, and oxidation), thermal (hydrothermal, and steam explosion) and biological (enzymatic treatment) methods [221,222,223]. Brief descriptions of each method have been provided in Table 3.

High-pressure homogenization is a common cell wall disruption method in industrial production, but this method can only deal with low concentrations of microalgae biomass. This method involves high energy and water consumption, and increases the cost of subsequent separation and purification of target compounds. High-pressure homogenization method improves the lipid recovery by up to 30% at a relatively lower temperatures (47 °C) [253]. Bead grinding method breaks cell walls and releases intracellular substances through high-speed movement of medium together with the grinding beads in the grinding cavity to generate various mechanical stress effects, such as collision, extrusion and shear forces [254]. A better disruption efficiency achieved by the bead milling results in the formation of larger lipid droplets compared to the high-pressure homogenization method and facilitates the downstream recovery of oil. A significant drawback of these two methods is the release of free fatty acids, decreasing oil stability [254].

Emerging green technologies such as high-intensity pulsed electric field, microwave, ultrasound, and supercritical fluid extraction have been developed as more powerful techniques for microalgae cell wall disruption [244,255]. Using high-intensity pulsed electric fields is a nonthermal technique of electroporation that alters the structure of cell membranes and walls. Externally applied electrical fields induce pores in the cell wall and thus improves lipid extractability [256,257]. Microwave technology is a thermal process wherein water and other polar molecules vibrate in the electromagnetic field to generate energy, which heats up the culture medium. The increase in intracellular temperature results in a pressure increase on the cell wall and induces the microalgae cell disruption [222,256]. When dealing with a small number of samples, the ultrasonic method has the advantages of simple operation and suitability for combination with other methods. In ultrasound pretreatment, microalgae are exposed to high ultrasonic waves, which produce cavitation bubbles around the microalgae cells. When the bubbles collapse, the induced shockwaves disrupt the cell walls and release or increase accessibility to the intracellular components, including lipids [122,257]. It has been demonstrated that this method is powerful in terms of releasing cellular components in contrast to bead milling, microwave and homogenization methods [258]. However, the ultrasonic method requires a lot of power if used to process a large number of microalgal cells or a large quantity of biomass. In addition, due to the uneven energy distribution, the ultrasonic method is generally limited to laboratory scale and has not been applied in the industrial production.

Biological pretreatment (antibiotics, enzymes and phage) has also been used for microalgae cell disruption [222,259]. The conditions of enzymatic method are mild, but this method can achieve a high cell-wall breaking rate and a low pollution rate. The most commonly used enzymes include snailase, trypsin, cellulose, lysozyme, β-glucanase, glucosidase, chitinase, endopeptidase, mannanase and proteases, and the lipid yield after enzymatic assisted extraction ranges between 7% and 85% depending on the enzymes, extraction methods, and microalgae species [3]. Considering the mechanisms of enzymatic reaction, the multilayered structure of microalgal cell wall imposes an additional challenge, requiring an enzyme cocktail. The microalgae cell structure property requires multi-stage pretreatments and complicated enzymatic cocktails, thus making it costly. These factors limit the application of enzymatic methods in the industry scale pretreatment of microalgae.

The selection of a method for microalgal cell-wall disruption requires the considerations of species due to different cell wall structure, cost and efficiency of methods, damage to the target biomolecules, environmental impact, and yield or recovery of the final products of interest (i.e., lipids). Accordingly, several alternative methods have been developed that are more cost-effective and protective of the target compounds, such as n-3 PUFA, and thus, their nutritional values, biological functions, and health benefits.

### 4.5. Extraction of Microalgae Oil

Extraction is critical to lipid production using microalgae [260,261]. After cell-wall disruption, oil extraction can be carried out using two different methods: supercritical fluid extraction and solvent extraction [51,262,263,264,265]. The principle of supercritical fluid extraction is to dissolve raw materials in supercritical gas, then adjust the temperature and pressure so that the solubility of different components in the raw material is changed, improving the efficiency of separation and extraction. Studies have shown that the concentration of PUFA increases while saturated fatty acids decrease with the increase in pressure. Therefore, supercritical fluid extraction methods can be used not only to extract lipids but also to achieve preliminary separation of fatty acids. In the solvent extraction method, microalgae raw materials are extracted first with a solvent mix, and then the matrix is separated into two phases by adding another solvent or changing the proportion of each solvent in the solvent mix. This results in the separation of the target compounds into one of two phases so as to achieve the purpose of preliminary separation and purification [223,266,267,268]. Chloroform is one of the most efficient solvents in terms of yield; however, it does not meet the food grade (Europe) or GRAS (USA) and cannot be used for the extraction of lipids that are targeted for food applications. The substitution with less efficient but GRAS solvents, such as ethanol, isopropanol, or hexane, is thus essential in the food industry [257]. Moreover, solvents are flammable and generally recognized as unhealthy [122], and thus, solventless extraction methods have been developed in recent years; for example, a solvent-free method has been developed to extract lipids from wet *N. oculate* [262]. In this method, oil is separated from aqueous phase by a saline solution combined with centrifugation. However, the oil yield is lower than the conventional solvent extraction method. To solve this problem, a super-critical extraction method is developed based on green solvents, but it operates in a high-pressure machine [215,222]. This method is costly and energy intensive; however, it offers a high selectivity for acylglycerols and minimizes co-extraction of polar lipids and nonacylglycerol neutral lipids. The selectivity to specific acylglycerols can be guided using polar modifiers [269].

Supercritical fluid extraction is up to five times faster than the solvent extraction [270]. Supercritical CO_2_ extraction seems to be the most adapted method for the extraction of high-value compounds such as n-3 PUFA, especially for meeting a high quality requirement of the final product. This technology has the advantages of a high extraction rate, no damage to PUFA, no solvent residue, no adverse impact to product smell, and being environmentally green. Furthermore, supercritical CO_2_ extraction has several advantages, including but not limited to, great stability, safety, operational convenience, low energy consumption, and low cost in the long term [222].

### 4.6. Concentration and Purification of Microalgae Oil

It is often difficult to obtain pure microalgae oil from extraction alone. Typically, concentration is performed via supercritical fluid equipment, liquid–liquid extraction, molecular distillation, urea fractionation, membrane extraction, precipitation and crystallization at low temperature, and others [271,272,273]. These industrial methods can be used to produce n-3 fatty acids in ethyl ester form with a purity of 90–95%. In the concentrating methods of methyl ester and ethyl ester, the oil is first esterified then separated and purified by structural differences between the target and other fatty acids, such as the degree of unsaturation and carbon chain length. In the urea complexation, fatty acids and urea are fully mixed and crystallized at a certain temperature; saturated fatty acids and monounsaturated fatty acids form inclusion complexes with urea while PUFA remain in the solution [274,275]. The freeze crystallization method, also termed “winterization” in the industrial practice, exploits the principle that different fatty acids differ in their solubility with or without organic solvents under low temperature conditions. This method is easy to operate, and the active ingredients are not prone to deterioration. The heavy metal complexation method is based on a property that metal ions such as silver salts can form polar complexes with carbon–carbon double bonds in unsaturated fatty acids. This method is costly and heavy metal contamination blocks this method from being used in food processing [276,277]. Supercritical fluid chromatography is a chromatographic method using supercritical fluid as the mobile phase. This method takes advantage of both supercritical fluid extraction and liquid chromatography [278,279].

Purification is the critical stage of the downstream processing of the desired products [280]. Microalgal oil after extraction contains cell debris, protein, and carbohydrates [281,282,283]. During the extraction process, impurities from raw materials and residues from the added reagents, such as alcohol, glycerol, water and catalyst remained after the reactions are mixed or left in the oil [271]. Molecular distillation has been applied to purify oil and yielded 98% segregation at 120 °C of evaporator temperature [284]. Free fatty acids are usually removed by chemical refining, such as alkalization. Bleaching uses typically absorbent clay or activated carbon to remove color pigments, oxidized products and trace metals. The bleached oil is then de-waxed to improve its clarity. A high-pressure steam is added to oil under high vacuum to remove the remaining oil components that contribute to taste, odor and color [100]. Every method used for oil purification has its advantages and disadvantages, and a single method cannot achieve a high purity requirement for human consumption. A solution is thus the use of multiple methods that can be combined to meet the industrial or commercial requirements for the purity of the final oil products.

## 5. Protection of PUFA via Microencapsulation

As mentioned in the previous sections, human consumption of PUFA, especially n-3 EPA and DHA, is insufficient worldwide [285]. Accordingly, food fortification and nutritional supplementation have been developed and used to address this issue. Multiple double bonds in PUFA molecules are unstable during food processing and storage due to their susceptibility to oxidation reactions, leading to the deterioration of product quality. Microencapsulation technologies can build multiple layers of wall materials around the dispersed small/fine oil droplets and effectively protect fatty acids from oxidation, inhibiting the generation of off-flavors and odors and improving the physicochemical functionalities, stability and bioavailability of fatty acids [286,287]. The standard microencapsulation technologies include spray drying, spray cooling/chilling, freeze drying, complex coacervation, fluidized bed coating, liposome entrapment, extrusion, and coextrusion [288,289].

### 5.1. Spray Drying

Spray drying is the most commonly used technique for the encapsulation of oils, which has the advantage of producing microcapsules through a relatively simple, continuous and inexpensive process, compared to other microencapsulation technologies [290]. This technology has been successfully employed for several decades by the food industry to encapsulate oils rich in PUFA [291]. The process involves the atomization of emulsions into a gaseous hot drying chamber, resulting in fast water evaporation and the formation of solid particles, i.e., oil droplets enveloped by solidified matrices (Figure 2) [290]. The commonly used wall materials in spray drying include proteins, carbohydrates, and gums, which are used either alone or in combination to achieve desired encapsulation efficiency and storage stability [292]. Carbohydrates and gums are the most widely used wall materials in the food industry simply because they are natural products, relatively inexpensive, and comparatively effortless in the acquirement of food regulatory approval. Proteins have excellent functionalities of film forming, gelation, foaming, emulsification and water holding capacity [293,294]. Due to their amphiphilic nature, proteins are natural emulsifiers that can reach the oil/water interface to form a physical barrier. Gelatin has been the first choice for coacervates-based encapsulation due to its biocompatibility, biodegradability, water retention ability and film formation ability [293,295]. Researchers from academia and industry are continually searching for an alternative to mammalian gelatin (porcine and bovine) due to socio-cultural and health-related concerns [296]. Plant proteins are less allergenic than their animal-derived counterparts [295,297]. A review on 14 research works published between 2013 and 2016 on the spray-drying encapsulation of omega-3-6-9 fatty acids-rich oils using protein-based emulsions concluded that plant proteins are gaining wider attention in recent years, in line with the consumer awareness and demands for “green” products [298]. Various emulsion systems can be used for the microencapsulation of oily or oil-soluble ingredients in the food industry including single-layered oil-in-water emulsion, multiple emulsion, and multilayered oil-in-water emulsion [290]. For the encapsulation of n-3 PUFA-rich oils, the oil-in-water emulsion is commonly used [293,299]. The droplet size and emulsion stability can be altered based on the characteristics of the wall materials, including the molecular weight, concentration, and emulsifying capability as well as other conditions such as solvent properties, pH, salt concentration, temperature, loading levels, and homogenization conditions [300,301,302,303,304,305,306,307,308,309]. 

Although there are several advantages, some drawbacks are linked to microcapsules prepared using spray-drying technology. Since the spray-drying process in the food industry is typically carried out using aqueous formulations, shell materials must have sufficient water solubility [290,310,311]. The fine powder produced in the spray-drying process can expose an explosion hazard to the surrounding areas and needs to be safely managed. In addition, high temperatures involved in the spray-drying process result in the oxidation of unsaturated fatty acids, especially PUFA, and other biomolecules that are sensitive to oxidation, consequently compromising the storage stability of oils [290].

### 5.2. Spray Cooling/Chilling (or Prilling)

Spray cooling is the process of solidifying an atomized liquid spray into particles. It is also referred to as spray chilling, spray congealing, or prilling. This process is often used to coat solid particles in a stream of cold gas to form micron-sized melt droplets, and the common matrix materials include fats, waxes, lipids, and gelling hydrocolloids [310,312]. Microspheres are the most common encapsulation morphology prepared with this technique, with an active ingredient dispersed homogeneously throughout the encapsulating matrix. The key difference between spray cooling and spray drying processes is that the former relies on cooling to solidify the final particles while the latter uses hot air to remove water to form dry particles. For the spray-chilling, the melting temperature of lipophilic materials is generally in a range of 34–42 °C, while for spray-cooling, the melting temperature is higher [313]. These two processes can be combined for double encapsulation of sensitive core ingredients. For instance, in a study, algal oil containing DHA was entrapped within a soy protein/sugar matrix with spray drying to produce microparticles around 80 μm, followed by spray chilling in a wax matrix to yield a particle size of 157 μm. The double encapsulation is more compact and can effectively prevent the oxidation of the core material [314].

### 5.3. Freeze Drying

Spray freeze-drying (SFD) overcomes the limitations associated with spray drying in the microencapsulation process. The SFD technique is a three-step operation: (1) spray the feed into droplets using an atomizer, (2) freeze the droplets with a freezing medium, and (3) sublime the water away from the droplets to a dried powder in a freeze dryer [315]. One unique advantage of this technique is that the processing temperature can be as low as that of the cryogenic liquid [316]. Therefore, it has been frequently used for thermos-sensitive core materials [317]. Studies have shown that the SFD powders have the appearance of higher porosity, which may result in enhanced oxygen penetration during storage and compromise the oxidative stability [290]. On the other hand, this feature may offer a higher release of active ingredients [318]. It is worth noting that SFD has a higher energy consumption, a longer processing time, and a high cost, which are disadvantageous compared to other drying methods [319].

### 5.4. Complex Coacervation

Complex coacervation is a phase-separation process in which the attractive electrostatic interactions drive oppositely charged biopolymers to form tiny aggregated colloidal particles [320,321]. Complex coacervation process generally consists of four steps (Figure 3), which are emulsification, coacervation, cooling/gelation, and solidification [322,323]. Among the various proteins and polysaccharides, gelatin and Arabic gum are the most widely studied and a pair of wall materials are used in the complex coacervation [293,295,297,324,325]. Nowadays, there is a trend to replace animal proteins with plant proteins in microencapsulating PUFA-rich oils [322]; soybean and pea protein isolates are the most commonly used plant proteins [326,327,328,329]. A formation of complex coacervation using chia seed protein isolate and chia seed gum has also been explored [330]. The same group has also used CPI-CSG (Chia seed protein isolate and chia seed gum) complex coacervates to microencapsulate PUFA-rich oils [330,331]. Complex coacervation technology has a higher embedding rate and can effectively protect ingredients that are senstive to oxidation, offering better emulsification and stability of the target comounds, although its process is complex and involves strict control point requirements and a high cost.

### 5.5. Nanoemulsions and Self-Emulsifying Emulsions

Nanoemulsions and self-emulsifying emulsions have become increasingly attractive in recent years, partly due to the demand for novel delivery vehicles to enhance the bioavailability of bioactive compounds [332,333,334]. Nanoemulsions have higher colloidal stability against gravity than the conventional emulsions [335]. The nano-sized droplets scatter less light, make the products transparent or translucent, and can be used to develop fortified beverages, soups, and sauces [334], with greatly improved bioavailability [336]. An oil-in-water nanoemulsion product was prepared using flaxseed oil that is rich in n-3 PUFA and different wall materials (alginate-whey protein/whey protein-sodium alginate) [336]. A study examined the oxidative stability of n-3 PUFA nanoemulsions, prepared by ultrasound using natural and synthetic emulsifiers, for a storage time of 5 weeks at 4 °C, 20 °C, and 40 °C, respectively [337]. The results of this study showed that Tween 40 is better than lecithin as an emulsifier in improving the oxidative stability of PUFA in oil-in-water nanoemulsions. In another study on the antioxidant and antibacterial activities of n-3 PUFA-rich oil nanoemulsions loaded in chitosan and alginate-based microbeads, the addition of natural antioxidant curcumin enhanced the encapsulation efficiency, loading capacity, and antioxidant activity of the formulated microbeads [338]. 

In recent years, a self-nanoemulsifying system has received increasing attention in n-3 PUFA delivery [339]. This system combines n-3 PUFA-rich oil, surfactant, cosurfactant/solvents, and other active components in a concentrated oil phase. Upon mixing into an aqueous solution, it spontaneously forms nanoemulsions by thermodynamic entropy gain and Gibbs-free energy reduction, resulting in the formation of fine droplets (<200 nm) [340,341,342]. The self-nanoemulsifying delivery system increases the product solubility, enhances the dissolution and improves the stability and bioavailability of the oil [341]. It has been reported that a self-nanoemulsifying emulsion formulation of DHA-rich oil improved the bioavailability and therapeutic efficacy of DHA and other PUFA [343]. Evidence is emerging that nanoemulsion is a promising technology to deliver n-3 PUFA-rich oil in foods, aiming at protecting the oil from oxidation, masking undesirable smells and flavors, and improving bioavailability [337,344,345].

### 5.6. Liposome

Liposome technology has been successfully applied in the pharmaceutical industry as a unique drug-delivery tool and is now finding its way into the food industry. Liposomes are spherical or nearly spherical vesicles with a bilayer membrane structure of various forms composed of different phospholipids. In a product preparation, n-3 PUFA-rich oil successfully embedded into liposomes through membrane hydration combined with ultrasound assistance [346]. In another study, carboxymethyl chitosan-coated n-3 PUFA-rich oil (fish oil extracted from Nile tilapia viscera) nanoliposomes, prepared using thin-film hydration combined with ultrasound, showed a better oxidative stability than uncoated liposomes. This finding suggests that the carboxymethyl chitosan layer probably had an inhibitory effect on the liposome decomposition and might have performed as a “shield” on the surface of liposome because of its stability in water [347].

## 6. Conclusions

Microalgae are a huge kingdom of microorganisms. Although research is falling behind in the understanding of lipids and fatty acid content and composition, especially n-3 PUFA in different species and strains, it has been well recognized that many microalgae species contain high contents of lipids and n-3 PUFA, and more importantly, the highly demanded EPA and DHA. The supply of n-3 PUFA from fish has increasingly been a challenge due to over-harvesting from marine sources and climate change. Aquaculture has been blooming in recent decades to address the shortage of marine sources; however, this industry sector requires a significant amount of n-3 PUFA-rich oils, which in fact compete with pets and humans. In seeking out alternative sources of n-3 PUFA, microalgae have been emerging as strategic crops because of their superior sustainability, environmental impact, high productivity, and abilities to synthesize and accumulate contents of oil as well as other high-value biomolecules, including but not limited to antioxidants. Thus, microalgae have been used commercially to produce n-3 PUFA, particularly EPA and/or DHA due to their significant interactions with growing conditions or environmental factors and accessibility for the application of bioengineering technologies. The biggest change for the time being is still the cost of production and processing. As such, microalgae are currently used mainly for the production of high-value EPA and DHA, which are applied predominantly in infant formulae, medicinal foods, and foods for the special population groups. There are a large number of species that are currently underexploited for their lipid profiles and production rates as a biological system to produce massive n-3 PUFA-rich oils. In addition, the harvesting and downstream processing require more research and technology/equipment developments to reduce the final cost. Although microcapsule technology can improve the bioavailability and prevent the degradation or deterioration of n-3 PUFA caused by oxidation, it is warranted to continue with the creation of new knowledge and the development of new technologies with significantly improved efficiencies. It is believed that with the advances in research on the biochemical composition of new species, the optimization of growing conditions, the development and application of bioengineering techniques and tools, the development of cultivation technology and facility, and the advances in product processing and delivery, the efficiency and profit of microalgae as a platform for commercial production of n-3 PUFA-rich oils will be substantially improved, contributing to the expected wide use of n-3 PUFA as an important functional ingredient for human consumption via foods or dietary supplements because of the long-recognized health benefits.

## Figures and Tables

**Figure 1 foods-11-01215-f001:**
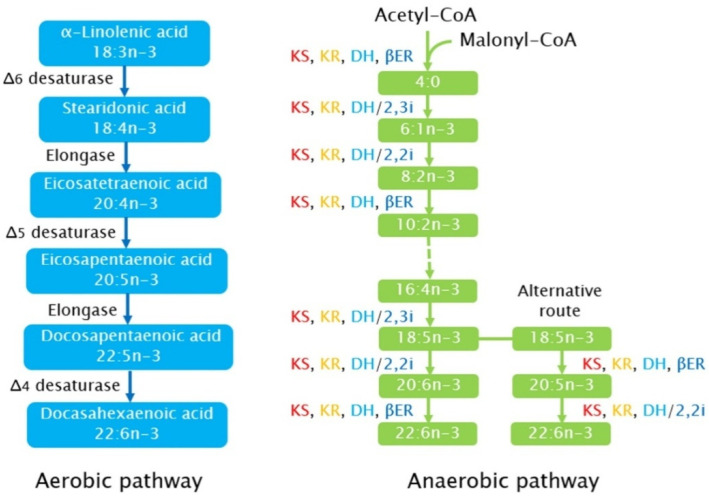
A schematic illustration of the aerobic and anaerobic pathways assumed for microalgae. For the anaerobic pathway, enzymes involved are the 3-ketoacyl synthase (KS—red), 3-ketoacyl reductase (KR—orange), dehydrase/isomerases (DH—light blue), DH/2,2i = dehydrase 2-*trans*, 2-*cis* isomerase, DH/2,3i = dehydrase 2-*trans*, 3-*cis* isomerase, and enoyl reductase (βER—blue). (Modified from figures in [1,10,78].

**Figure 2 foods-11-01215-f002:**
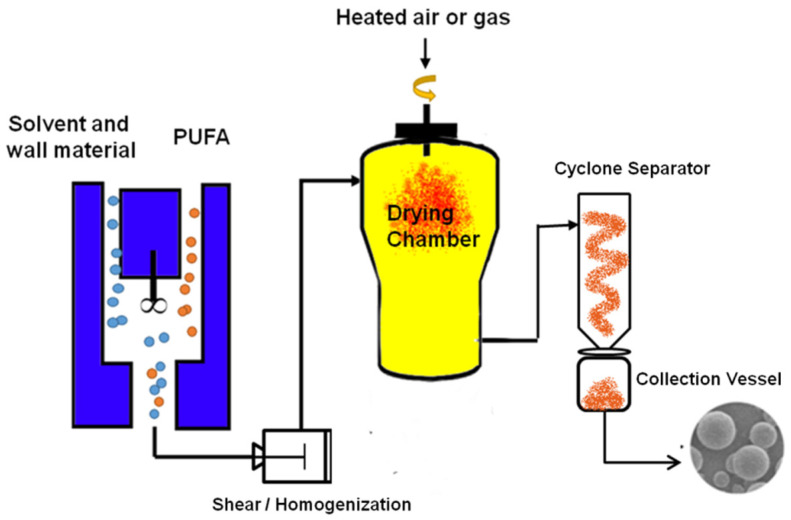
The process of spray drying.

**Figure 3 foods-11-01215-f003:**
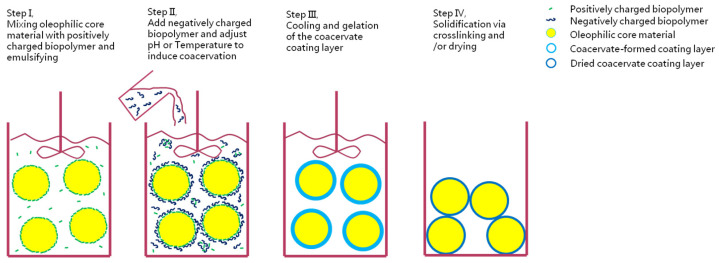
The process of complex coacervation.

**Table 1 foods-11-01215-t001:** Fatty acid composition (% of total fatty acids) of different microalgae species.

Species	14:0	16:0	16:1	18:0	18:1	18:2	18:3	20:5 EPA	22:6 DHA	Total (%DM)	Ref.
	% of Total Fatty Acids										
*Aurantiochytrium*	2.9	39.8		0.5	0.5	0.1	0.4	0.5	46.7	40–55	[52,53]
*Chlamydomonas reinhardtii*		4–20	3.8		1–16	1–10	2–22		0–5.4	12–64	[54]
*Crypthecodiniumcohnii*	18	12–45		3	8				13–55	25–63	[53,55,56]
*Dunaliella* sp.		10–28	12–16		8–11	5.9	12–36		14–21	12–46	[36,53]
*Emiliana huxleyi*	18.9	10.3		10.8	19.8				9.2		[41]
*Euglena gracilis*	0.9	11.3	1.3		3.1	3.5	19.3		9.0		[41]
*Heterococcus chodati*		10.0	30.6			8.1		32.6			[41]
*Nannochloropsis oculata*	4.2	14–24	24–30		3–5	2.9	0–9	27–49		22–37	[36,37]
*Pavlova lutheri*	9.7	20.1	26.3		1.7	0.5	0.4	18.2	9.8	35	[41]
*Phaeodactylum tricornutum*	4.4	14–16	40–60		8.1	1.0	20–30	18.4	1.4	32	[57]
*Scenedesmus obliquus*		30.7		23.3	6–25	8–18	10–33			21–58	[58]
*Schizochytrium*	2–8	20–45		4.8	38.4	7.9	1.2	5–12	5–50	51–71	[53,59,60]
*Thraustochytrium* sp.	1.6	16.8		0.2	0.2	0.2		7.5	69	13	[52]
*Tribonema vulgare*	4.1	13.3	34.4				10.5	17.4			[41]
*Ulkenia* sp.		25–30			10–12			5–15	15–30	20–52	[52]

**Table 2 foods-11-01215-t002:** Harvesting method of microalgae cells.

Method	Description	Advantage	Disadvantage	Example	Ref.
Sedimentation	Natural gravity sedimentation relies on the particle size of microalgae cells and the density difference of culture environment to harvest; suitable for large biomass and fast sedimentation rate.	Simple; Inexpensive	Affected by cell morphology, not applicable to small-diameter and low-density algae	The filamentous *Spirulina platensis* having a sedimentation velocity of 0.64 m/h.	[201]
				The diatom *Amphora* having a velocity of 2.91 m/h.	[202]
				*Monoraphidium* sp. can be harvested after 24 h with a yield of 98%.	[203]
Coagulation- Flocculation	Coagulation and flocculation employ chemical (coagulant, zeta potential and pH) or physicochemical (e.g., hydrodynamics) principles to promote cell aggregation and form large particles for separation purposes	Efficient; Inexpensive	Possible coagulant contamination	At high pH, Fe^3+^, Ca^2+^ and Mg^2+^ induced coagulation of *C. reinhardtii* at <5 mM with >90% biomass harvesting efficiency.	[204]
				Adjusted pH to 9.5 induced coagulation of *Chaetoceros calcitrans* with 89% of cells were harvested.	[205]
Centrifugation	Centrifugal method uses acceleration to harvest cells. Various types of centrifugal equipment can be used to harvest microalgae, such as spiral plate centrifuge, decanter centrifuge, disk stack centrifuge, and hydrocyclone.	Efficient; No chemical pollution	High energy consumption; Expensive; Affected by algae morphology	A low biomass harvest efficiency of approx. 50% at 9000× *g* for *Helical A. platensis* filaments.	[206]
				A harvest efficiency of 99.3% achieved at 3000× *g* for 10 min for *S. obliquus* cells.	[207]
Flotation	Flotation is a method to transfer microalgae to the surface of culture medium by introducing bubbles (air or ozone), and then collect microalgae by skimming.	Efficient	High energy consumption	Using 3.8 L flotation cell and dissolved air flotation, the harvest efficiency reaches 91%.	[208]
				The heat-induced flotation of *Scenedesmus dimorphus* at 85 °C, with a harvest efficiency around 80%.	[209]
Membrane filtration	Membrane filtration can be employed as dead-end or tangential flow filtration mode with membrane pore size varied from 0.1 μm to 10 μm for microfiltration and a few nanometers to 0.1 μm for ultrafiltration membrane respectively.	Pollution-free	Easy to be corroded by medium; Blockages need to be cleaned	Driven by gravity, *A. platensis* cultures collected using 5 μm nylon membrane with over 90% harvest rate.	[206]
				In the harvesting of *Arthrospira* sp. with ceramic microfiltration and ultrafiltration membranes, fluxes of 230 L m^−2^ h^−1^ and 93 L m^−2^ h^−1^ reported, respectively.	[210]
Drying	The water content of microalgae can be reduced to 10%. There are many drying methods, such as sun-drying, freeze-drying, oven-drying, spray drying, and drum drying.	Lower moisture content; Efficient	Long time; High energy consumption; Uneconomical (except sun drying)	Sun drying is done under sunlight, usually at 18–27 °C; the efficiency is 400–1200 mmol m^−^^2^ s^−^^1^; takes 2–3 days.	[211]
				Oven drying is done using hot air, usually at 60 °C, takes 12 h.	[211]

**Table 3 foods-11-01215-t003:** Overview of major microalgae cell disruption methods.

Method	Description	Advantage	Disadvantage	Ref.
Chemical Method				
Hydrothermal	Hydrothermal pretreatment is based on cell wall rupture due to internal pressure build-up from the heating, and hydrolysis of cell wall components by steam explosion, autoclave and water bath treatment.	Unrestricted moisture content; Suitable for low value targets; No chemical reagent; Simple operation	High temperature may oxidize and degrade lipids and other bioactives; High energy consumption	[224,225]
Acid/Alkaline treatment	Inorganic acid or alkaline solution is used to catalyze and promote hydrolysis processes as an improved version of hydrothermal pretreatment	Efficient; Simple operation	Enhance the soluble chemical oxygen demand; Degradation of sensitive compounds	[226,227]
Oxidative pretreatment	Strong oxidant (such as ozone or hydrogen peroxide)is used togenerate hydroxyl radicals (OH^-^) that attack and disrupt the cell walls of microalgae.	Efficient; Suitable for the preparation of biofuels	Destroy highly oxidizable compounds	[228,229,230]
**Physical method**				
Pressing	A mechanical force is used to demolish the thick membrane of microalgae and release the oil content. Screw press, extruder, and biomass spraying are the main means of the mechanical pressing.	High purity of the target products; No chemical pollution	Require highdryness of the biomass	[231]
Bead beating	The membrane of microalgae is disrupted by the action of fast-moving spinning beads.	Simple equipment; Efficient; Wide application range	Need cooling equipment; high temperature destabilize target compounds; Emulsification of products	[232]
High-pressure homogenization (HPH)	HPH is typically used for emulsification but is also suitable for a large-scale disruption of microalgae cells.	Efficient; High biomass concentration; Reduction of viscosity	Product emulsification affects subsequent extraction	[233,234,235]
Ultrasonication	Highpressure bubbles and their cavitation generate shock waves, producing high shear forces.	Simple; Suitable for combination with other methods	Oxidation target product; Affect fatty acid chain length; Low efficiency	[236,237]
Pulsed Electric Field	An intense electric field for very short durations (pulses)applied to microalgae cells to induces reversible or irreversible pores creation (electroporation) on the cell membranes to aid their disruption.	Suitable for freshwater microalgae; Gentle	Low efficiency; Additional steps to remove salt (cost up); Not applicable to marine microalgae	[238,239,240]
**Other novel pretreatment methods**				
Enzymatic methods	It is a specific pretreatment method, and requires high selectivity of suitable enzymeson the cell wall structure and composition of a special typeof algae.	High specificity; Mild reaction conditions; Low energy consumption	High enzyme cost; Short process time	[232,241,242]
CO_2_ explosion	It pressurizes CO_2_ inside the cell and increases intracellular gas concentration, leading to excessive expansion and cell rupture. Other non-reactive gasses such as N_2_ are also used.	Prevent degradation of target products; Efficient	High-cost	[243,244]
Electricity-based methods	High voltage electric discharges (HVED) utilizes electrodes of needle-plate geometry to deliver high voltage pulses to microalgae suspensions. HVED additionally induces thermal and mechanical effects to the cells due to cavitation and shockwave formation. Non-thermal plasma is another electricity-based method where a needle-to-plate electrode geometry is placed in an argon filled reactor.	No chemical pollution; High extraction rate	Not suitable for extraction of unsaturated fatty acids	[233,245,246]
Osmotic shock	A week-long pretreatment in which microalgae cells are broken up due to the density difference between cytoplasm and high salt solution.	Simple; High extraction rate	Time consuming	[247,248]
Ionic liquids	Ionic liquids form a large number of hydrogen bonds that interact with polymers such as cellulose, and destroy the original hydrogen bonds in cellulose and break the cell wall.	High extraction rate; Room temperature	Loss of ions over time; Potential ions pollution	[249,250]
Viral cell lysis	Virus-assisted cell disruption is a novel method that appeals for low energy consumption.	No chemical pollution	Unknown control factors	[251,252]

## Data Availability

Not applicable.

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
