# Peer review of "Production, Processing, and Protection of Microalgal n-3 PUFA-Rich Oil"

_foods, 2022, doi:10.3390/foods11091215_

Round 1

Reviewer 1 Report

This article represents an extensive and fairly complete review of the use of microalgae for the production of oils rich in polyunsaturated fatty acids of the n-3 series. The work focuses not only on the biochemical details of the production of fatty acids by different microalgae, but also gives a fairly complete overview of the different aspects that are important from a practical point of view. In this sense, it reviews the different environmental factors that influence production and the technologies available for the extraction, separation and protection of the fatty acids produced.

Author Response

The authors thank you for your comments and support.

Reviewer 2 Report

The article titled:"Production, Processing, and Protection of microalgal n-3 PUFA-rich oil" is well written and includes many intersting and important information. It deals with an urgent topic of biosynthesis pathways and engineering processes that can be applied to more efficient production of n-3 PUFA. The Authors also discussed  the impact of environment on n-3 PUFA production and the methods that allow extraction and purification of oil and techniques that can be applied to protect n-3 PUFA from oxidation.

The article is organized properly, all important information are gathered and thoroughly described, but there is no single scheme, picture, diagram. In my opinion, the review article should include schemes, pictures, diagrams to visualize what is intended to be described. Authors should add some schemes. Additionally, please complete the chapter dealing with the possibilities of potential application of microalgal n-3 PUFA-rich oil and its microencapsulated analogues in food industry. The information should be added, if such products have been already used in food industry.

Author Response

I do appreciate your comments! Please see the attachment.

Reviewer 3 Report

Thank you very much for allowing me to evaluate your work. The article is interesting and analyses the topic of study in depth.

It is necessary to include a methodology section specifying how the literature search was conducted and whether it is systematic or narrative.

The references of the 100-106 should be added.

Lines 452-455 should go into more detail on the new methodologies mentioned.

Author Response

(The authors gave the same response as above.)

Round 2

Reviewer 2 Report

I would like to thank the Authors for improving the article according to Reviewer's comments and recommendations.

Please, check the correctness of Figure 3 presentation.